# Geometric and Information Compression of Representations in Deep Learning

## Abstract

Deep neural networks transform input data into latent representations that support a wide range of downstream tasks. These representations can be characterized along information-theoretic and geometric dimensions, but their relationship remains poorly understood. A central open question is whether low mutual information (MI) between inputs and representations necessarily implies geometrically compressed latent spaces and vice versa. We investigate this question using neural collapse as a measure of geometric compression and theoretically sound MI estimation in conditional entropy bottleneck (CEB) networks and continuous dropout networks. We evaluate the interplay between MI, geometric compression, and generalization on classification tasks under controlled noise injection schemes. Our findings show that low MI does not reliably correspond to geometric compression, and that the connection between the two is more nuanced than often assumed. We conjecture that generalization acts as a potential confounder in this connection rather than being a direct consequence.

## 1 Introduction

The quest for understanding generalization in deep neural networks (DNNs) inspires many researchers to propose different approaches to this important problem (Keskar et al., 2016; Dziugaite & Roy, 2017; Jiang et al., 2019; Liang et al., 2019; Petzka et al., 2021). It is intuitively evident that the ability to generalize depends on how the DNN transforms input data into latent representations at hidden layers, i.e., on the properties of latent representations. With the rise of foundation models, research on representation properties has gained increasing attention, as their effectiveness heavily depends on the characteristics of the latent space.

There have been efforts to analyze the geometrical properties of latent representations, for example, by investigating the manifold formed by them. Following Occam's razor, for classification problems it is natural to seek representations that lie on separated, low-dimensional manifolds associated with different classes. Low intrinsic dimension as an estimate of the manifold dimension in latent space has been suggested to correlate with good generalization performance (Blier & Ollivier, 2018). Among the proposed ways to capture geometric compression, one of the most actively studied is neural collapse, which characterizes the class-specific clustering of latent representations and its connection to generalization (Papyan et al., 2020).

Taking a probabilistic point of view where the input data is drawn from a distribution, one can denote latent representation as a random variable with distribution implicitly described by the DNN. Again following Occam's razor, the information bottleneck (IB) theory favors representations that have a large mutual information (MI) with the target but small MI with the input. Compressed MI with inputs indicates that irrelevant input details are discarded and overfitting is avoided. While training with the standard cross-entropy loss ensures that the MI between the latent representation and the target is large, it was claimed (and later disputed by Saxe et al. (2018)) that stochastic gradient descent implicitly reduces the MI between latent representation and input (Shwartz-Ziv & Tishby, 2017).

Prior work has suggested a close connection between clustering of representations and information-theoretic compression as defined by the IB principle. Goldfeld (2019) investigated estimates of MI in DNNs with additive Gaussian noise and concluded that a reduction of MI throughout training is correlated with tightening of the clusters of latent representations. Geiger (2022) further argues that many MI estimators are inherently geometric. In this work, we contribute to this debate and investigate

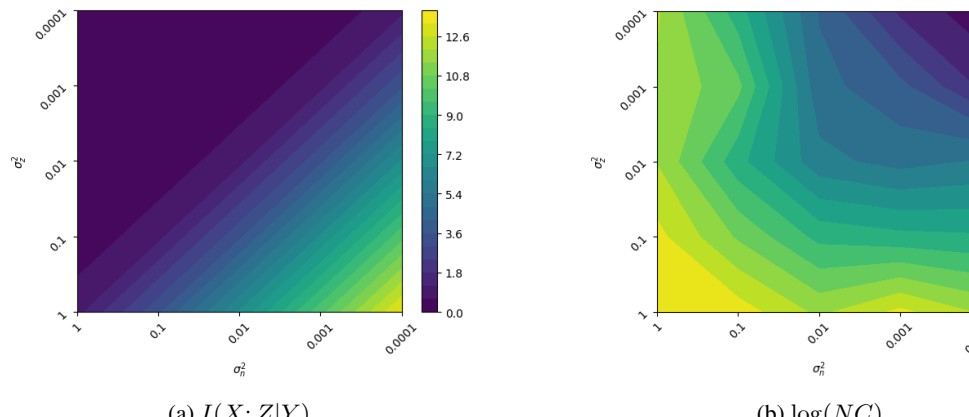

(a) $I(X; Z|Y)$          (b) $\log(NC)$

Figure 1: Toy model illustrating the interplay between information-theoretic and geometric compression in terms of neural collapse. Data points $x \in X$ with class labels $y \in Y$ are encoded into latent representations $Z \sim \mathcal{N}(\mu(x), \sigma_n^2 I)$, where class centroids $\mu_y \sim \mathcal{N}(0, I)$ are perturbed by encoder spread $\sigma_z^2$ and noise variance $\sigma_n^2$. We vary $\sigma_z^2$ and $\sigma_n^2$ and estimate geometric compression via NC, averaged over 50 trials. $I(X; Z|Y)$ is evaluated analytically. Results show that low MI arises either from strong noise ($\sigma_n^2$ large) or from tightly clustered encodings ($\sigma_z^2$ small), while NC indicates compression only in the latter case. This demonstrates why low MI and NC do not have to coincide. For the details on the toy example see Appendix D.

the interplay between geometric and information-theoretic compression using previously unexplored empirical setups, also conjecturing a link to generalization. In Section 3 we describe the setups used for empirical evaluation: DNNs trained with conditional entropy bottleneck (CEB) (Fischer, 2020), i.e., models with additive data-dependent noise, and Gaussian dropout DNNs, i.e., models with multiplicative fixed noise. For both types of models we estimate MI with the input (Section 3.1) and compute a measure for characterizing neural collapse (Section 3.2). In contrast to information plane analysis (Saxe et al., 2018; Goldfeld, 2019) which tracks the behavior of MI throughout training, we are interested in the end-of-training values. This way we analyze the interplay between MI compression, neural collapse, and model performance.

Our contributions are as follows:

1. We show that DNNs with continuous dropout form a test bed for information-theoretic analyses by proving that the MI between input and latent representations is always finite (Theorem 3.1);

2. We perform large-scale analysis estimating MI between inputs and representations in state-of-the-art neural networks;

3. We perform detailed investigations of the connection between information compression and geometric compression in terms of label-specific clustering.

In Section 4 we show with 8 architectures (including fully connected DNNs, convolutional DNNs, and transformers) and around 500 models trained on 5 different datasets (including vision and natural language processing tasks) that the connection between geometric and information-theoretic compression is not as simple as initially suggested (cf. Figure 1). In fact, we find a negative non-linear correlation between the two. Moreover, varying hyperparameters of training, we can force the correlation to become positive, which suggests that there is an unknown confounder for the relation between geometric and information compression. We conjecture that this confounder is related to generalization abilities.

## 2 RELATED WORK

The connection between representations compression and generalization is explored in multiple research works. Shwartz-Ziv & Tishby (2017) claim that compression of MI between input and latent

representation is the implicit regularization of stochastic gradient descent and see it as an explanation of generalization abilities of DNNs. In its turn, in DNNs with additive noise (Goldfeld, 2019), the tightening of class clusters coincides with a reduction in MI between inputs and representations over the course of training. Similarly, Patel & Shwartz-Ziv (2024) show that the local rank of the representations reduces throughout training and connect this to the Gaussian information bottleneck (Chechik et al., 2003). These works thus suggest a tight link between geometric and information-theoretic compression, which in its turn leads to good generalization. Indeed, for DNNs with additive noise it was shown that neural collapse leads to both grokking and to compression in the sense of IB Sakamoto & Sato (2025). Although one might expect that tighter clusters of representations always correspond to reduced MI, this intuition is misleading. Cluster tightness and MI quantify different properties, and our numerical toy example (Figure 1) demonstrates that clusters can become more compact without any decrease in MI. More generally, compressibility in a geometric sense is used by algorithms that extract low-dimensional manifolds while preserving relevant information, such as Globerson & Tishby (2003) or Marx & Fischer (2022), which indicates that reducing geometrical dimension does not necessarily lead to information reduction. Geiger (2022) thus attribute the link between MI and geometric compression to the inherently geometric properties of some MI estimators, while noting that the connection to generalization remains unclear. In particular, it is established that estimating MI using popular binning estimators directly reduces the estimate to a measure of geometric compression, cf. (Geiger, 2022, Fig. 2).

In contrast to the aforementioned studies that follow MI and clustering throughout training, end-of-training studies that focus on representations after convergence of the model and allow to investigate the connection to generalization in a clearer way. For instance, Skean et al. (2024) analyze kernel-based estimates of entropy of language model representations and find that intermediate layers exhibit lower entropy and higher downstream task performance, linking entropy compression to generalization. Cheng et al. (2023) observe a positive correlation between perplexity and the intrinsic dimension of last-token representations in transformers and connect both measures to the performance of finetuning. None of the previous works embark on estimating MI between inputs and representations, but use some approximations of the amount of information in the representation.

**Geometric Characterization of the Representation Space** A central challenge in studying geometric compression is the lack of a clear definition. As a result, most works operationalize geometric compression through proxies, such as representations' clustering structure in classification tasks. Empirically, it has been observed that the penultimate layer representations of well-trained, state-of-the-art classification DNNs collapse such that each class maps to a single point, with these points arranged at the vertices of a simplex equiangular tight frame. This tight class-wise clustering was termed neural collapse (Papyan et al., 2020) and is a sufficient condition for generalization since it implies linear separability. However, it is not a necessary condition (Han et al., 2025). As an example such DNNs as RevNets which are reversible (Gomez et al., 2017) or Parseval models with orthogonality enforced on weights (Cisse et al., 2017), have very limited capabilities for neural collapse, but still achieve state-of-the-art generalization performance. To date, efforts to relate information-theoretic compression to neural collapse have not yielded conclusive results.

**Mutual Information in DNNs** Estimating MI between input data and representations in deterministic models with continuous distributions is provably vacuous, since this MI is infinite (Amjad & Geiger, 2019). One of the ways to address this issue is to modify the representations to be stochastic in some way, e.g., by injecting noise. There are two ways of injecting noise: additive and multiplicative. These schemes can be implemented either with noise of fixed variance or variance that depends on the input data point. So, Goldfeld (2019) estimated the MI by adding Gaussian noise with fixed variance to each neuron output. The deep variational IB method (Alemi et al., 2017) and its variants (Fischer, 2020) add random noise to the neuron outputs, with the mean and variance learned from the input data. This setup has not previously been analyzed from the perspective of information compression, a gap our paper addresses. Adilova et al. (2023) analyzed multiplicative noise with both fixed and adaptive variance. While adaptive variance follows the information dropout framework of Achille & Soatto (2018) and ensures finite MI estimates, fixed-variance dropout was introduced there as a novel setup. As a theoretical foundation, they proved that continuous dropout guarantees finite MI between inputs and representations in DNNs with ReLU activations. In this work, we extend this result to arbitrary activation functions, thereby enabling information-theoretic analysis across modern architectures and supporting large-scale empirical evaluations.

## 3 METHODOLOGY

We consider a supervised classification setting, that is, DNNs trained on a dataset $\mathcal{S}$ sampled from an unknown (typically continuous) distribution $\mathcal{D}$ on $\mathcal{X} \times \mathcal{Y}$. Here, $\mathcal{X} \subseteq \mathbb{R}^n$ is the space of inputs and $\mathcal{Y} = \{1, \ldots, c\}$ is the set of $c$ classes. We denote by $X$ the multivariate random variable that describes the input to the DNN, and by $Z$ some representation of $X$ that it produces. The unknown distribution $\mathcal{D}$ and the DNN jointly induce a distribution of $Z$ in the representation space $\mathcal{Z} \subseteq \mathbb{R}^d$.

We consider two types of models in this work: DNNs trained with the Conditional Entropy Bottleneck (CEB) (Fischer, 2020), which implements data-dependent additive noise, and models regularized with Gaussian dropout, which implements fixed multiplicative noise. We chose these models because they complement the existing literature: Models with fixed additive noise were considered by Goldfeld (2019), while data-dependent multiplicative noise was examined by Adilova et al. (2023).

CEB trained models are slight modification of the variational information bottleneck (VIB) models, with only difference in the distribution at the bottleneck layer [1]. Variational IB models have demonstrated effectiveness across applications including multi-view learning (Federici et al., 2020), multi-task learning (Qian et al., 2020), and invariant representation learning (Razeghi et al., 2023; Moyer et al., 2018), illustrating the practical relevance of this modeling framework. Models trained with the CEB objective consist of a deterministic decoder and a stochastic encoder that draws $Z$ from a Gaussian distribution with a mean vector and a covariance matrix that depends on the respective input $x \in \mathcal{X}$. Equivalently, the encoder can be assumed to map the input $x$ deterministically via a learned function $f$ and then add zero-mean Gaussian noise $D(x)$ with a data-dependent covariance, i.e., $Z = f(x) + D(x), D(x) \sim \mathcal{N}(0, \sigma^2(x)I)$. Assuming that $(x, y)$ are realizations of a pair of random variables $(X, Y)$, the CEB training objective is given as:

$$\mathcal{L}_{CEB} = I(X; Z|Y) + \beta L_{ce}(f(X) + D(X), Y) \ , \tag{1}$$

where $L_{ce}$ is the cross-entropy loss and $\beta$ trades between classification performance and compression of MI. CEB-trained models thus explicitly compress MI between input and latent representation $I(X; Z|Y)$. As long as $\beta$ is finite, the learned noise variance $\sigma^2(x)$ will be positive for every $x$ due to regularization; as a consequence the $I(X; Z|Y)$ is always finite.

Multiplicative noise via dropout represents a widely used form of stochasticity, present in nearly all state-of-the-art architectures. Although continuous Gaussian dropout has not been as widely adopted as its Bernoulli counterpart, its effect on training is similar and in some aspects more advantageous, as noted in prior work (Srivastava et al., 2014). Importantly, we establish that continuous multiplicative noise renders MI between inputs and representations finite and quantifiable, providing a theoretically sound basis for information-theoretic analysis of dropout-regularized networks within a frequentist framework. The latent representations $Z$ equal $f(X) \circ D$, where $\circ$ denotes element-wise multiplication, and $D \sim \mathcal{N}(1, \sigma^2 I)$. For these models it was shown that $I(X; Z)$ is finite at least if $f$ is a DNN with ReLU activation functions, cf. Theorem 3.3 and Proposition 3.5 of (Adilova et al., 2023). The following theorem extends this result to real analytic activation functions, which include common activation functions such as sigmoid or tanh. With this result, we effectively show that any DNN with continuous dropout has a provably finite MI between inputs and representations. This creates the possibility to employ the information-theoretic analysis framework for any state-of-the-art frequentest DNN without resorting to purely stochastic or quantized models.

**Theorem 3.1** (Mutual Information is Finite in Continuous Dropout Networks). *Consider a non-zero deterministic DNN function $f : \mathbb{R}^n \to \mathbb{R}^d$ constructed with finitely many layers, a finite number of neurons per layer, and non-constant real analytic activation functions. Let $X$ be a continuously distributed RV with bounded probability density function $p(x)$ and bounded support.*
*Let $Z = f(X) \circ D(X)$, where $D(X) = (D_1(X), \ldots, D_N(X))$ is (potentially data-dependent) noise with components conditionally independent given $X$ such that all $D_i(X)$ have essentially bounded differential entropy and second moments, i.e., $\mathbb{E}\left[D_i(X)^2\right] \leq M < \infty$ $X$-almost surely for some $M$. Then, $I(X; Z) < \infty$.*

**Proof sketch** In order to prove this theorem for analytical activation functions one needs to provide a proof that $\mathbb{E}_X \log|f(X)| = \int_K \log|f(x)| \, p(x) \, dx$ is finite (see (Adilova et al., 2023)). Since $p(x)$

---

[1]The main difference between CEB and variational IB (Alemi et al., 2017) is that the variational marginal for $Z$ is parameterized by the class label. In other words, while variational IB models $Z$ as a Gaussian, CEB models $Z$ as a Gaussian mixture with number of components corresponding to number of classes.

is bounded and $K$ is compact, the integral can only diverge to $-\infty$ near zeros of $f$. Because $f$ is real analytic, its zero set $Z$ has a structured form: by the Lojasiewicz Structure Theorem, $Z$ is a finite union of lower-dimensional varieties (of dimension at most $n-1$). Hence $Z$ has measure zero and the integral is well-defined. Near the zeros, the Lojasiewicz inequality for analytic functions ensures that $|f(x)|$ is bounded below by a polynomial in the distance to $Z$: $|f(x)| \geq C \cdot \text{dist}(x, Z)^q$. Thus, $\log |f(x)|$ is controlled by $\log(\text{dist}(x, Z))$ near $Z$. Finally, by stratifying $Z$ into smooth manifolds $M_j$ of various dimensions and integrating in polar coordinates around each $M_j$, the local integrals reduce to terms of the form $\int_0^\epsilon \log(r)\, r^m \, \mathrm{d}r$, which are finite for any $m \geq 0$. Summing over all components gives the result. The full proof is provided in Appendix B.

It should be noted that any latent representation that follows the dropout layer also has a finite MI with input due to the data processing inequality.

### 3.1 Estimating Mutual Information

With finite MI between $X$ and $Z$ guaranteed, the challenge is to estimate it from a dataset $\mathcal{S} \sim \mathcal{D}$.

**CEB** For CEB-trained models, the variational bound on $I(X; Z|Y)$ is directly embedded in the training objective, so the MI estimate is available "for free" during training. By definition, the variational formulation of equation 1 is

$$\min_{q_{Z|Y}} \mathbb{E}\left[ KL(e_{Z|X}(\cdot|X) \| q_{Z|Y}(\cdot|Y)) \right] + \beta L_{ce}(f(X) + D(X), Y) \ , \tag{2}$$

where the encoder $e_{Z|X} = \mathcal{N}(f(x), \sigma(x))$ is defined by the DNN and where the expectation is taken w.r.t. the data distribution $\mathcal{D}$. In this equation $\mathbb{E}\left[ KL(e_{Z|X}(\cdot|X) \| q_{Z|Y}(\cdot|Y)) \right] = I(X; Z|Y) + \mathbb{E}\left[ KL(p_{Z|Y}(\cdot|Y) \| q_{Z|Y}(\cdot|Y)) \right]$ (cf. (Geiger & Fischer, 2020, eq. (12a))). However, this bound estimate of MI is only useful if the gap $\mathbb{E}[KL(p_{Z|Y} \| q_{Z|Y})]$ remains small. In the CEB setting, both the encoder $e_{Z|X}$ and the variational distribution $q_{Z|Y}$ are optimized jointly, which actively reduces this gap. This co-training ensures that the variational expression provides not just a formal bound, but also a practically tight surrogate for $I(X; Z|Y)$. As a result, the CEB loss can be interpreted as a reliable estimate of conditional MI rather than a loose upper bound.

**Dropout** In contrast to CEB, models trained with Gaussian dropout require an MI estimator. We evaluated multiple state-of-the-art MI estimators and selected the Difference of Entropies (DoE) estimator (McAllester & Stratos, 2020) as the most stable under the conditions of measuring MI for high-dimensional continuously distributed vectors. To estimate MI between input $X$ and representation $Z$, the DoE estimator computes $I(X; Y) = H(Z) - H(Z|X)$ by separately estimating the unconditional entropy $H(Z)$ and the conditional entropy $H(Z|X)$. In practice, $H(Z)$ is approximated using a parametric marginal distribution $q_Z$, typically a simple Gaussian or logistic distribution with learnable parameters. The conditional entropy $H(Z|X)$ is approximated using a neural network $q_{Z|X}$ that predicts a distribution over $Z$ conditioned on $X$. Both terms are estimated via the mean negative log-likelihood (cross-entropy) of observed samples:

$$H(Z) \approx -\frac{1}{N} \sum_{i=1}^{N} \log q_Z(y_i) \ ,$$

$$H(Z|X) \approx -\frac{1}{N} \sum_{i=1}^{N} \log q_{Z|X}(z_i|x_i) \ .$$

During training, we minimize the negative log-likelihoods of both approximations. Iteratively, this process makes the learned distributions $q_Z$ and $q_{Z|X}$ as close as possible to the true marginal $p_Z$ and conditional $p_{Z|X}$, respectively. The MI is then obtained as the difference between the two estimates. A key advantage of DoE is that it provides a stable, finite-sample estimate of MI, particularly when the true MI is large, which often leads to saturated estimates with traditional variational lower bounds.

## 3.2 MEASURING GEOMETRIC COMPRESSION

We characterize geometric compression via the unified neural collapse characteristic measure suggested by Galanti et al. (2021). Specifically, we compute:

$$\text{NC} = \frac{1}{c(c-1)} \sum_{i=1}^{c} \sum_{j=i+1}^{c} \frac{\text{Var}_i + \text{Var}_j}{\|\mu_i - \mu_j\|^2} \tag{3}$$

where $\mu_i = \sum_\ell z_\ell / K_i$ and $\text{Var}_i = \sum_\ell \|\mu_i - z_\ell\|^2 / K_i$ are the mean and variance of $K_i$ latent representations sampled from inputs $X$ belonging to class $i$, and where $\|\cdot\|^2$ is the squared Euclidean norm. Formally, this reflects only NC1 property of Neural Collapse and therefore we look only at the clustering properties of representation. It is more reliable than Sihloette score or binned entropy in the high dimensions and also widely used as characteristic of geoemtric properties of representation space (Galanti et al., 2021; Han et al., 2025)

## 3.3 REGULARIZATION

To investigate the relationship between mutual information and class-wise clustering strength, we train models under varying regularization settings. These settings are chosen to induce different training behaviors while still maintaining reasonable performance.

For CEB-trained models, we vary the $\beta$ coefficient in Equation 1 to control the strength of conditional MI compression between inputs and latent representations. Strong compression (small $\beta$) increases the cross-entropy loss, impairing the model's ability to learn and generalize, whereas weaker compression allows more information to flow and focuses optimization on cross-entropy. The resulting set of models exhibits varying generalization and provides a controlled testbed for examining whether geometric and information-theoretic compressions are correlated under additive adaptive noise.

A natural way to obtain a set of differently generalizing models with continuous dropout would be to vary the dropout variance; however, in practice this is not viable, as only a narrow range of dropout values allows for stable and efficient training. To address this, we introduce a regularizer on clustering tightness, based on NC in Equation 3, which is subtracted from the cross-entropy loss to prevent clusters from becoming too tight. The regularizer takes the form $\lambda(\tanh(\alpha\text{NC}))$, where $\lambda$ controls the strength of regularization, $\alpha$ scales NC into the effective range of $\tanh(\cdot)$, and $\tanh(\cdot)$ ensures boundedness and saturation of the regularization term contribution. This approach produces a range of models with varying degrees of NC and correspondingly different generalization abilities.

## 4 EXPERIMENTS

In this section, we present the experimental setup and evaluate geometric and information compression in two settings: additive adaptive noise in CEB-trained models and multiplicative constant noise in Gaussian dropout models. For dedicated latent variables $Z$, we quantify clustering using NC introduced in Equation 3 and estimate the MI between inputs and $Z$ using either the variational bound implicit in the CEB loss or the DoE estimator discussed in Section 3.1. We then analyze the correlation between these measures and generalization and performance of the models.

Since nonlinear relationships are expected to exist between these quantities, we use the rank correlation instead of the linear (Pearson) correlation. Moreover, different datasets and architectures lead to different ranges of the considered quantities, and normalization is required. Specifically, we first convert the quantities in each experimental setup into ranks, normalize these ranks, and then evaluate the linear correlation of the collection of ranks over all experiments. For details on the ranking computation see Appendix C.1.

## 4.1 CEB-TRAINED NETWORKS

We devised four setups of the experiments with CEB. In each setup, we swept $\beta$ over $1000, 500, 100, 50, 10, 5, 2$ and trained five models per value with different random seeds. We observed that very small $\beta$ leads to underfitted models, while getting $\beta$ too large can lead to overfitting.

| | Measures | Rank correlation | |
|---|---|---|---|
| | | Train | Test |
| $\beta$ **dep.** | acc $\|\beta$ | 0.9 | 0.88 |
| | gen $\|\beta$ | 0.21 | - |
| | MI $\|\beta$ | 0.76 | 0.84 |
| | NC $\|\beta$ | -0.99 | -0.98 |
| | MI $\|$ NC | **-0.75** | **-0.83** |
| **Perf.** | acc $\|$ MI | 0.59 | 0.74 |
| | acc $\|$ NC | -0.89 | -0.88 |
| | gen $\|$ MI | 0.33 | - |
| | gen $\|$ NC | -0.21 | - |

Table 1: Correlations for CEB experiments. Each of the 4 setups with a different network-dataset combination is trained with 7 values of $\beta$, 5 random seeds each. The correlation is computed between ranked values. *acc* stands for accuracy, *gen* for generalization gap in terms of accuracy.

| | Measures | Rank correlation | |
|---|---|---|---|
| | | Train | Test |
| $\lambda$ **dep.** | acc $\|\lambda$ | -0.02 | -0.3 |
| | gen $\|\lambda$ | 0.09 | - |
| | MI $\|\lambda$ | -0.48 | -0.25 |
| | NC $\|\lambda$ | 0.96 | 0.94 |
| | MI $\|$ NC | **-0.46** | **-0.24** |
| **Perf.** | acc $\|$ MI | -0.17 | 0.07 |
| | acc $\|$ NC | -0.05 | -0.34 |
| | gen $\|$ MI | 0.09 | - |
| | gen $\|$ NC | 0.07 | - |

Table 2: Correlations for Gaussian dropout experiments. 4 setups with different network-dataset combination is ran with 5 random seeds for each of the $\lambda$ in a grid of different values. The correlation is computed between ranked values. *acc* stands for accuracy on the train or test data correspondingly, *gen* for generalization gap in terms of accuracy.

The four considered setups are as following:

1. An MLP (5 hidden layers with 1024 neurons each, ReLU activation functions) as an encoder for FashionMNIST. Dimensionality of $Z$ is selected to be 256. The decoder is an MLP with one hidden layer of size 1024 and ReLU.

2. LeNet5 (LeCun et al., 1998) encoder for FashionMNIST. Dimensionality of $Z$ is selected to be 64. The decoder is an MLP with one hidden layer of size 1024 and ReLU.

3. WideResNet-28-4 (Zagoruyko & Komodakis, 2016) encoder for CIFAR-10. Dimensionality of $Z$ is selected to be 256. Decoder is an MLP with one hidden layer of size 1024 and ReLU.

4. Densenet121 (Huang et al., 2017) encoder for CIFAR-100. Dimensionality of $Z$ is selected to be 256. Decoder is an MLP without hidden layers.

The backward encoder, i.e., the variational distribution $q_{Z|Y}$, only has to emulate the parameters of a Gaussian mixture model, therefore in all the setups it is a shallow MLP without any hidden layers. In CEB-trained models, the latent representation is stochastic by design, which requires sampling in order to evaluate both the training loss and downstream measures: In each setup we drew 8 samples per data point from the representation distribution during training. The same sampling procedure was applied at evaluation time, and MI was extracted from the CEB loss. For details on the training setup see the Appendix C.2.

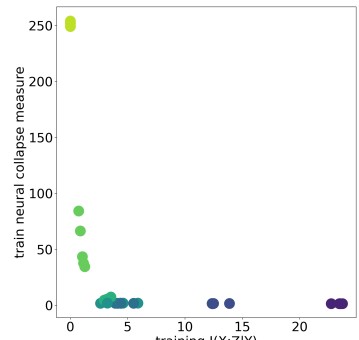

Figure 2: NC against MI on the train data for CEB (DenseNet121 on CIFAR-100). Strong clustering can correspond to large and small amount of information, same like strong compression of MI can correspond both to weak and strong clustering.

In Table 1 we present the correlations for the whole experiment. We analyze correlations between the generalization gap and the metrics of MI and NC using their values on the training set. This is because only measures derived from the training data can meaningfully indicate how well the model will perform on unseen test data. In all four of the setups we see a clear monotonic and non-linear correlation between the NC measure and conditional MI $I(X; Z|Y)$, both for test and train data. As an exemplary setup we also demonstrate this non-linear correlation in the experiments

with DenseNet121 in Figure 2. It is obvious that large MI can be consistent with strong clustering of the representations, and vice-versa. We refer to an analysis of this phenomenon in Figure 1, where we demonstrate that information-theoretic compression can be achieved by both clustering equivalent to small NC and by uninformative, noisy representations with large NC. At the same time, strong clustering can still lead to large MI values if the encoder noise is sufficiently small. Thus, the interplay between information-theoretic and geometric compression in a sense of clustering is quite intricate. In the first group of rank correlations in Table 1, we demonstrate that larger $\beta$ leads to higher classification accuracy and larger MI, which is expected due to the CEB training objective. $\beta$ is negatively correlated with the NC measure, which means larger $\beta$ leads to representations that are clustered. Indeed, geometric compression is an indicator of good performance in terms of accuracy: The more clustered the representations with respect to labels, the higher the accuracy. The correlation between accuracy and MI is positive, but not as pronounced. This is expected from the CEB training objective, as $\beta$ shifts optimization efforts between compression of MI and performance (see eq. 1).

## 4.2 Gaussian Dropout Neural Networks

We devised four setups for the experiments with multiplicative noise. For the experiments we fix the dropout variance found through hyperparameter search that allows to achieve close to the state-of-the-art training results. We apply the NC regularization described in Section 3.3 and in one setup we vary the regularization strength $\lambda$ to a subset of values in the grid from $-50$ to $50$. We also include the setup without any regularization ($\lambda = 0$). Each of $\lambda$ runs was repeated with 5 different random seeds. We employ standard neural architectures, but in each integrate at least one Gaussian dropout layer and add a fully connected layer of dimensionality 128 or 512 as the penultimate layer corresponding to the hidden variable $Z$.[2] In order to estimate MI between inputs and representation we employ the logistic DoE estimator from McAllester & Stratos (2020) as discussed in Section 3.1. We sample representations from the trained model without switching off the dropout for estimating MI. We obtain 4 different representations for each of the inputs and use this data to train the estimator. In the CEB experiments only conditional MI is available, but in this setup estimating conditional MI is impossible reliably due to the limited amount of samples in every class. Nevertheless, due to Markov chain property of $Y \to X \to Z$ we have that $I(X; Z|Y) = I(X; Z) - I(Z; Y)$, where $I(Z; Y)$ is bounded above by $\log K$, where $K$ is the number of classes. $I(X; Z)$ in our experiments is several orders of magnitude larger than $\log K$, therefore, using $I(X; Z)$ rather than $I(X; Z|Y)$ preserves the relative ordering of models, which is the only information required for correlation analysis. For the information on the estimator architecture and its training we refer the reader to the Appendix C.4.

The different setups of the experiment are as follows:

1. ResNet-18 (He et al., 2016) trained on CIFAR-10. For this model we added the dropout layer with variance $0.4$ before the fully-connected layer at the end of the architecture and representation layer of dimensionality 128.

2. VGG11 (Simonyan, 2014) with batch normalization trained on SVHN. For VGG11 we added a dropout layer with variance of $0.5$ in the classifier module after the first layer and representation layer of dimensionality 512.

3. Densenet121 (Huang et al., 2017) trained on CIFAR-100. In this architecture we replaced the original binary dropout with continuous Gaussian dropout with variance of $0.3$ and representation layer of dimensionality 128.

4. MiniBERT (Bhargava et al., 2021; Turc et al., 2019) trained on the AG's News Corpus dataset (Zhang et al., 2015). In this architecture we applied Gaussian dropout to the initial embedding layer with variance of $0.6$ and representation layer of dimensionality 512.

See Appendix C.2 for additional details on the training hyper parameters.

We display the correlations in the setup in Table 2. We again observe negative correlation between the MI and the neural collapse measure, showing that tighter clustering and compressed information do not coincide. The character of correlation is the same as in the CEB experiments. Differently from

---

[2]We introduce a layer of a dimensionality less than the usual final fully-connected layers in such architectures in order to speed up MI estimation and NC computation, but overall the experimental setup is possible without it as well.

CEB experiments, $\lambda$ does not give a direct control over classification accuracy and its correlation with MI compression is also much weaker. Only the obvious correlation to NC is very strong, and opposite to correlation with MI. The correlation of the measures, both NC and MI, with accuracy and generalization gap is also very weak.

### 4.3 OBSERVATIONS

The rank correlations in Tables 1, 2 in all experiments consistently indicate a negative relationship between class-wise clustering and MI compression. In the Gaussian dropout setup, however, this trend is weaker, as some training configurations showed little correlation or even a positive one. In Figure 3

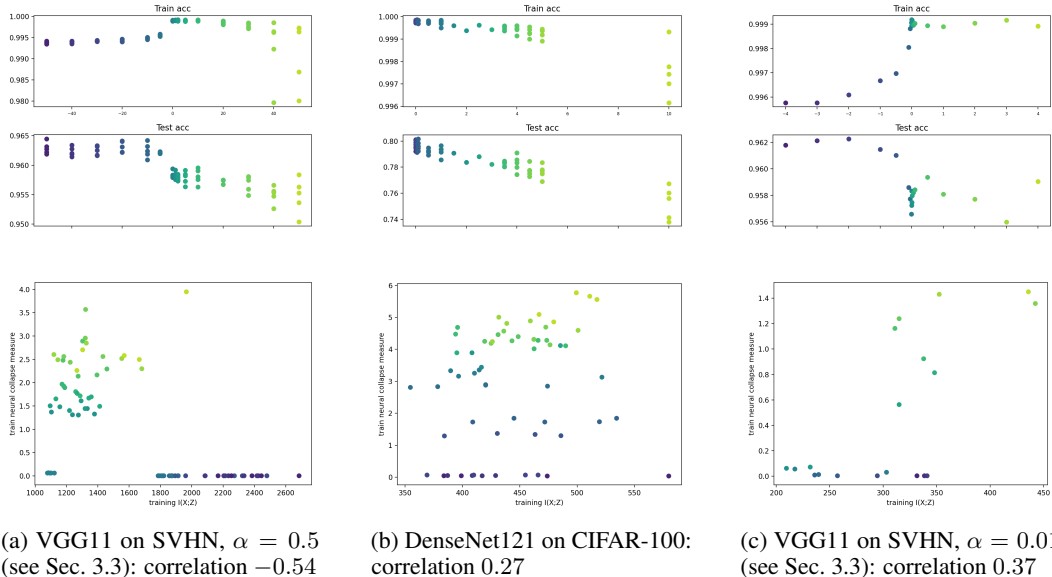

(a) VGG11 on SVHN, $\alpha = 0.5$ (see Sec. 3.3): correlation $-0.54$

(b) DenseNet121 on CIFAR-100: correlation $0.27$

(c) VGG11 on SVHN, $\alpha = 0.01$ (see Sec. 3.3): correlation $0.37$

Figure 3: Different hyperparameter setups can result in different correlation between MI and NC (second row), which is possibly confounded by generalization performance (first row).

we demonstrate the run on CIFAR-100 (second column) which has similar range of MI for very different values of NC and two different training regimes of VGG11, where one resulted in non-linear negative correlation and another in positive correlation. These results suggest that DNN training with the standard cross-entropy loss involves a complex interplay between geometry and information. In particular, our experiments reveal that strong clustering (low NC) can be associated with various levels of MI, disputing their simple relationship proposed in previous works. Our observations further indicate that generalization ability may act as a confounder in this relationship (Figure 3, first row), thereby calling into question the reliability of both measures as predictors of generalization.

## 5 DISCUSSION AND CONCLUSION

With this paper we contribute to the debate about the connection between geometric compression in terms of clustering of representations and information-theoretic compression in terms of the mutual information between latent representations and inputs. Previous research has claimed a positive (potentially non-linear) correlation between both types of compression: MI is small if latent representations are clustered, cf. (Goldfeld, 2019), and MI estimation is inherently geometric in many cases, cf. (Geiger, 2022). We performed experiments on networks with adaptive additive noise (variational IB) and (fixed) continuous multiplicative noise (Gaussian dropout); for the latter we provide a proof that the true MI is finite. Our empirical evaluation of Gaussian dropout networks demonstrates how an information-theoretic perspective can be applied to state-of-the-art DNNs, providing a foundation for future research. The experiments also reveal that the correlation between information compression and geometric compression is negative and nonlinear.

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

## A   VISUALIZATION OF LATENT REPRESENTATIONS

Plots in Figure 4 and Figure 5 demonstrate visually that the effect of clustering structure of the representation space on generalization is hard to describe. Visually tight and convex clusters in Figure 4 (a) result in worse performance than in (b). At the same moment enforcing weak clustering of representations in Figure 5 (b), which visually looks as more tight clusters, is outperformed by the model with clustering in (a).

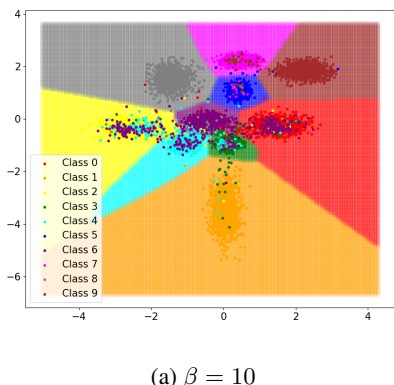
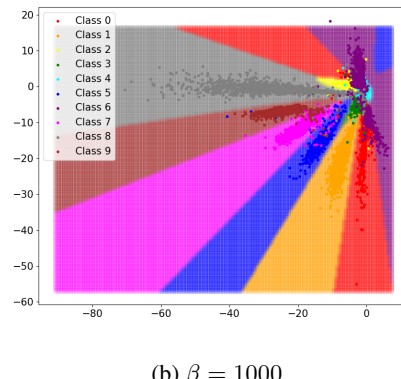

(a) $\beta = 10$          (b) $\beta = 1000$

Figure 4: Difference in the structure of clusters of representations extracted from a CEB model trained on FashionMNIST. The dimensionality of the representation is chosen to be 2 for visualization. Points correspond to the representations of the test data and the background color corresponds to the decision boundaries of the model. $\beta$ regulates the compression of the mutual information via loss regularization: The smaller it is, the stronger $I(X; Z|Y)$ is compressed.

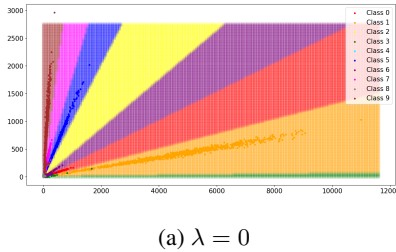
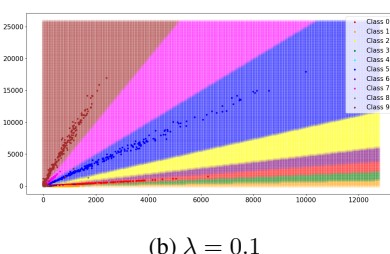

(a) $\lambda = 0$          (b) $\lambda = 0.1$

Figure 5: Difference in the clusters structure of the representations extracted from a fully connected network with Gaussian dropout with $p = 0.4$ trained on FashionMNIST. We let the dimensionality of the representation to be 2 for visualization. The points correspond to the representations of the test data and the background color corresponds to the decision boundaries of the model. $\lambda$ regulates the strength of clustering in representations: The larger it is, the weaker is the clustering.

## B   PROOF OF THEOREM 3.1

To prove the theorem, we use the following proposition, which extends Proposition 3.5 of (Adilova et al., 2023) to non-constant real analytic activation functions. The proof then directly follows from Theorem 3.3 in (Adilova et al., 2023).

**Proposition B.1.** *Consider a non-constant deterministic NN function $f\colon \mathcal{X} \to \mathbb{R}$ constructed with finitely many layers, a finite number of neurons per layer, and real analytic activation functions. Let $X$ be a continuously distributed RV with PDF $p_x$ that is bounded ($0 \leq p_x(x) < C_p$ for all $x \in \mathcal{X}$) and has support in a compact and connected set $K$. Then the conditional expectation $\mathbb{E}[\log(|f(X)|)]$ is finite.*

*Proof.* By the law of unconscious statistician, we have

$$\mathbb{E}_X\left[\log(|f(X)|)\right] = \int_K \log(|f(x)|)p(x)dx \tag{4}$$

As $p(x)$ is bounded and $K$ is bounded, it is clear that $\mathbb{E}_X\left[\log(|f(X)|)\right] < \infty$. It remains to investigate the behavior around $f(X) = 0$ to show $\mathbb{E}_X\left[\log(|f(X)|)\right] > -\infty$. Since $f$ is a composition of real analytic functions, $f$ is itself real analytic. We will make use of the well-behaved structure of real analytic functions to bound the expectation.

The idea of the proof is as follows: The set of zeros of a real analytic function is a set of measure zero. However, this does not suffice to guarantee finiteness of $\mathbb{E}_X\left[\log(|f(X)|)\right]$ as the behavior around these zeros matters. In one dimension, we can directly use the fact that an analytic function $f : \mathbb{R} \to \mathbb{R}$ locally behaves polynomially around zeros. In higher dimensions, an analytic function can remain 0 along submanifolds of lower dimension. The natural analog to the behavior in one dimension is that the function behaves polynomially as a function of the distance to the set of zeros. Once we know that $f$ behaves polynomially, we can use that $\int_0^\epsilon log(r)q(r)dr$ is finite for any polynomial $q(r)$.

Technically, we apply the Lojasiewicz's Structure Theorem (see Theorem 6.3.3 in Krantz & Parks (2012)), which states that the set of zeros of a non-constant real analytic function is a finite collection of algebraic varieties of dimension at most $n - 1$. In particular, the set of zeros is a null set and Equation 4 is well-defined. Let $Z_i, i = 1, \ldots, k$ denote the connected components of $Z$.

Moreover, the Lojasiewicz inequality (see Theorem 6.3.4 in Krantz & Parks (2012)), gives that around zeros of $f$, the function $f$ behaves polynomially in the distance to the zero: That is, let $Z$ be the set of zeros of $f$ and $dist(x, Z)$ the distance function from $x$ to $Z$. Let $U$ be an open neighborhood with non-zero intersection with $Z$. Then, for each compact subset $E$ of $U$, there is a constant $C_L > 0$ and a natural number $q$ such that

$$|f(x)| \geq C_L \cdot dist(x, Z)^q \tag{5}$$

for all $z$ in $E$.

Consider now $\epsilon$-neighborhoods $N_i^\epsilon, i = 1, \ldots k$ around the connected components $Z_i$, where $\epsilon$ is chosen such that $|f(x)| < 1$ for all $x \in N_i^\epsilon \cap K$ and all $i$. Then,

$$\mathbb{E}_X\left[\log(|f(X)|)\right] > \sum_i^{finite} \int_{N_i^\epsilon} \log(|f(x)|) \cdot p(x) \, dx + \int_{K \setminus \bigcup_i N_i^\epsilon} log(|f(x)|) \cdot p(x) \, dx \tag{6}$$

The last integral is bounded below by boundedness of $K$, continuity of $f$, and the fact that $x$ has at least distance $\epsilon > 0$ to any zero of $f$. It remains to show that each $\int_{N_i^\epsilon} \log(|f(x)|) \cdot p(x) \, dx > -\infty$. We fix $i$ in the following and remove it from the notation for readability.

We use the Lojasiewicz inequality equation 5 to find $C_L > 0$ and $q \in \mathbb{N}$ such that $|f(x)| \geq C_L \cdot dist(x, Z)^q$ on $N^\epsilon$.

By using Whitney stratification (Whitney, 1965), a real analytic algebraic variety can be decomposed into a finite number of connected smooth submanifolds, $Z = \bigcup_j^{finite} M_j$. Let $d_j$ denote the dimension of $M_j$. We know that $0 \leq d_j < n$.

Using the decomposition, we will need that $f$ locally behaves polynomially in distance to each $M_j$. Since Equation 5 only measures the distance to the variety $Z$, we will need to partition the space into regions where the distance to $Z$ is defined by the distance to $M_j$. For this, we let

$$N_j^\epsilon = \{x \in N^\epsilon \mid dist(x, M_j) = dist(x, Z)\} \ .$$

Then

$$\bigcup_j N_j^\epsilon = N^\epsilon$$

since every distance needs to be realized by at least one of the connected smooth manifold.

For each $j$, we introduce local coordinate systems $x^j = (x_1^j, \ldots, x_{d_j}^j)$ of the manifold $M_j$ and $x^{j,\perp} = (x_1^{j,\perp}, \ldots, x_{n-d_j}^{j,\perp})$ of its normal space. Now we calculate the integral:

$$\int_{N^\epsilon} log(|f(x)|)p(x)dx \geq \sum_{j}^{finite} \int_{N_j^\epsilon} log(|f(x)|) \cdot p(x)dx$$

$$\geq \sum_{j}^{finite} \int_{N_j^\epsilon} log(|f(x)|) \cdot C_p \, dx$$

$$\geq \sum_{j}^{finite} \int_{N_j^\epsilon} log(C_L \cdot dist(x,Z)^q) \cdot C_p \, dx$$

$$= \sum_{j}^{finite} \int_{N_j^\epsilon} \log(C_L \cdot dist(x,M_j)^q) \cdot C_p \, dx$$

$$\geq \sum_{j}^{finite} \int_{M_j} \int_{B_\epsilon^{n-d}(0)} \log(C_L \cdot dist(x,M_j)^q) \cdot C_p \, dx^\perp \, dx^j$$

$$= \sum_{j}^{finite} \int_{M_j} vol_{n-d_j-1}(S^{n-d_j-1}) \int_{r=0}^{\epsilon} \log(C_L \cdot r^q) \cdot C_p \, r^{n-d_j-1} \, dr \, dx^j$$

$$= \sum_{j}^{finite} C_p \cdot vol_{d_j}(M_j) \cdot vol_{n-d_j-1}(S^{n-d_j-1})$$

$$\cdot \left( \frac{\epsilon^{n-d_j}}{n-d_j} \log(C_L) + q \cdot \int_{r=0}^{\epsilon} log(r) \cdot r^{n-d_j-1} dr \right)$$

$$> -\infty.$$

The final inequality holds since $M_j$ lies in a bounded set and the last integral is finite for $n-d_j-1 \geq 0$, i.e., $d_j < n$. □

## C  EXPERIMENT DETAILS

### C.1  CORRELATION ESTIMATION

Mathematically, if $\{a_i^j\}$ and $\{b_i^j\}$, $i = 1, \ldots, N_j$, are quantities recorded in the $j$-th experiment by varying a certain parameter, let $r_{a,i}^j$ and $r_{a,i}^j$ denote the ranks of $a_i^j$ and $b_i^j$ such that $r_{a,i}^j = 1$ if $a_i^j > a_\ell^j$ for all $\ell \neq i$, etc. The ranks of the $j$-th experiment are then min-max normalized to the range of $[0,1]$, and the normalized ranks are collected in a vector $\tilde{r}_a = [\tilde{r}_{a,1}^1, \tilde{r}_{a,2}^1, \ldots, \tilde{r}_{a,N_1}^1, \tilde{r}_{a,1}^2, \ldots, \tilde{r}_{a,N_M}^M]$. We then report the Pearson correlation coefficients between $\tilde{r}_a$ and $\tilde{r}_b$.

### C.2  TRAINING HYPER PARAMETERS

For all the setups in CEB experiments we employ Adam optmizer with learning rate $1e - 3$. We trained models for $150$ epochs with exponential learning rate scheduler with $\gamma = 0.97$. Batch size is selected as following:

1. FC + FMNIST: $64$
2. LeNet5 + FMNIST: $64$
3. WRN28-4 + CIFAR-10: $256$
4. DenseNet-121 + CIFAR-100: $256$

For the setups in Gaussian dropout experiments the following hyper parameters were used:

1. ResNet18 + CIFAR-10: batch size $128$, learning rate $0.1$, training for $100$ epochs with SGD with weight decay of $5e - 4$, momentum $0.9$ and cosine annealing learning rate scheduler.

2. VGG11 + SVHN: batch size 256, learning rate 0.01, training for 150 epochs with SGD with weight decay of $5e - 4$, momentum 0.9 and cosine annealing learning rate scheduler.

3. DenseNet-121 + CIFAR-100: batch size 256, learning rate 0.1, training for 200 epochs with SGD with weight decay of $5e - 4$, momentum 0.9 and cosine annealing learning rate scheduler.

4. mini-BERT + AG News: batch size 256, learning rate $1e - 5$, training for 20 epochs with AdamW with weight decay of $1e - 2$ and cosine annealing learning rate scheduler.

### C.3  WHY DOE?

In this section we summarize the theoretical basis of DoE, explain why it yields a lower bound on mutual information, and describe the neural density models used to approximate the marginal and conditional distributions. We also include the role of the logistic-based DoE formulation and the construction of negative samples.

For any random variable $Z$ with true density $p(z)$, the entropy is

$$H(Z) = -\mathbb{E}_{p(z)}[\log p(z)].$$

DoE replaces the intractable $\log p(z)$ with the log-density of a neural estimator $q_\phi(z)$, trained by maximum likelihood on samples of $z$:

$$\hat{H}(Z) = -\mathbb{E}_{p(z)}[\log q_\phi(z)].$$

Using the standard decomposition,

$$-\mathbb{E}_{p(z)}[\log q_\phi(z)] = -\mathbb{E}_{p(z)}[\log p(z)] + \mathrm{KL}\big(p(z) \,\|\, q_\phi(z)\big),$$

we immediately obtain

$$\hat{H}(Z) = H(Z) + \mathrm{KL}(p(z) \,\|\, q_\phi(z)) \geq H(Z).$$

Thus, a variational estimator cannot underestimate entropy; it always overestimates it by the non-negative KL divergence.

The same argument applies to the conditional entropy:

$$\widehat{H}(Z|X) = -\mathbb{E}_{p(x,z)}[\log q_\theta(z|x)] = H(Z|X) + \mathbb{E}_{p(x)}\mathrm{KL}(p(z|x) \,\|\, q_\theta(z|x)).$$

Again, the estimate is an overestimate of the true conditional entropy.

Mutual information can be expressed with entropies as follows

$$I(X; Z) = H(Z) - H(Z|X).$$

DoE substitutes the variational surrogates:

$$\hat{I}_{\mathrm{DoE}} = \hat{H}(Z) - \widehat{H}(Z|X).$$

Substituting their decompositions yields

$$\hat{I}_{\mathrm{DoE}} = H(Z) + \mathrm{KL}(p(z)\|q_\phi) - \big(H(Z|X) + \mathbb{E}_{p(x)}\mathrm{KL}(p(z|x)\|q_\theta)\big),$$

$$= I(X; Z) - \Big(\mathbb{E}_{p(x)}\mathrm{KL}(p(z|x)\|q_\theta) - \mathrm{KL}(p(z)\|q_\phi)\Big).$$

The conditional KL is typically much larger than the marginal KL. No matter that $p(z \mid x)$ is usually a unimodal distribution, while $p(z)$ is a multimodal mixture over all inputs, estimating $H(Z \mid X)$ often requires a more expressive model because learning the mapping $(x, z) \mapsto$ conditional code length is estimator–wise more difficult than modeling the marginal $p(z)$. Hence,

$$\mathbb{E}_{p(x)}\mathrm{KL}(p(z|x)\|q_\theta) \gg \mathrm{KL}(p(z)\|q_\phi),$$

which implies

$$\hat{I}_{\mathrm{DoE}} \leq I(X; Z).$$

Thus DoE produces a lower bound on mutual information through upper bounds on corresponding entropies.

**Neural Density Models Used in DoE**   DoE models both the marginal and conditional densities with neural networks:

- A marginal density estimator $q_\phi(z)$, trained on samples $\{z_i\}$.

- A conditional density estimator $q_\theta(z|x)$, trained on pairs $\{(x_i, z_i)\}$.

The density estimator employed in the logistic-based DoE variant is a factorized logistic mixture model. The logistic parameterization is more flexible than a Gaussian model and avoids the assumption of unimodality, which is particularly important for $p(z|x)$.

For the "logistic" option, DoE models the marginal or conditional distribution of $Y \in \mathbb{R}^d$ as a diagonal Logistic distribution with parameters $\mu \in \mathbb{R}^d, s = \exp(\ln \mathrm{var}) \in \mathbb{R}^d_{>0}$. For a single dimension, the Logistic density is

$$p(y \mid \mu, s) = \frac{\exp\left(-\frac{y-\mu}{s}\right)}{s\left(1 + \exp\left(-\frac{y-\mu}{s}\right)\right)^2}.$$

Define the standardized variable $w = \frac{y-\mu}{s}$. The exact negative log-likelihood is

$$-\log p(y \mid \mu, s) = w + 2\log(1 + e^{-w}) + \log s.$$

In the vector case $(y, \mu, s \in \mathbb{R}^d)$, the code computes

$$\mathcal{L}(Y) = \frac{1}{B}\sum_{b=1}^{B}\sum_{i=1}^{d}\left[w_{b,i} + 2\log(1 + e^{-w_{b,i}}) + \log s_i\right],$$

where $B$ is the batch size. In the implementation we compute:

$$\mathtt{whitened} = (Y - \mu)/s \quad \longleftrightarrow \quad w,$$

$$\mathtt{adjust} = \mathrm{softplus}(-w) = \log\left(1 + e^{-w}\right),$$

and the returned value

$$\mathtt{negative\_ln\_prob} = \mathrm{mean}_b\left[\sum_{i=1}^{d}\left(w_{b,i} + 2\,\mathrm{softplus}(-w_{b,i}) + \ln s_i\right)\right]$$

is exactly the multidimensional negative log-likelihood of a factorized Logistic distribution.

**Construction of Negative Samples for Dropout Network**   The conditional model $q_\theta(z|x)$ is trained via a contrastive objective derived from noise-contrastive estimation. For each positive pair $(x_i, z_i)$, one constructs a set of negative samples $\{(x_i, z_j) \mid j \neq i\}$, with $z_j$ drawn from the empirical marginal distribution of features in the mini-batch. In practice:

- choose $z_i$ as the positive sample for $x_i$,

- use all other $\{z_j\}_{j \neq i}$ from the batch as negatives.

This encourages the logistic conditional model to assign higher likelihood to the true conditional distribution while pushing away unrelated feature vectors.

The resulting conditional log-likelihood objective is

$$\mathcal{L}_{\mathrm{cond}} = \sum_i \left[\log q_\theta(z_i|x_i) - \log \sum_j q_\theta(z_j|x_i)\right],$$

which is the logistic-form surrogate used in DoE. For each sample of the dataset we produce several noisy representations with dropout active for estimating the conditional entropy. We never pass noisy representations of one and the same input in one batch, not to violate the negative sampling (noisy representations of the same input are not negative samples of the distribution).

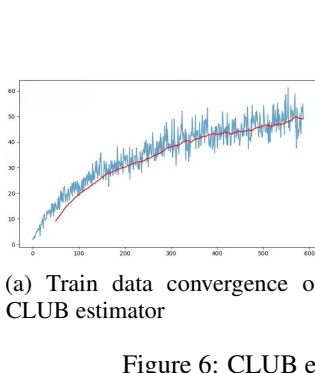

(a) Train data convergence of CLUB estimator

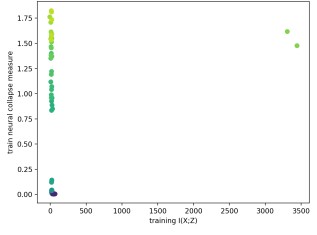

(b) Geometric compression VS MI measured by CLUB

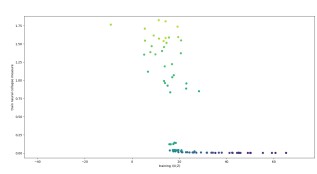

(c) Geometric compression VS MI measured by CLUB, without outliers

Figure 6: CLUB estimator on ResNet18 representations with CIFAR-10 data.

### C.3.1 MI ESTIMATORS COMPARISON

We estimated MI using MINE (Belghazi et al., 2018) and CLUB (Cheng et al., 2020) in one of the configurations, ResNet18 on CIFAR-10. With CLUB we obtained quite good convergence (Figure 6(a)) and a qualitatively similar picture to DoE except for two outliers (Figure 6(b)). It is visible still, that DoE achieves significantly better convergence (Figure 8(a)). With MINE the picture is more distorted (Figure 7(b)), but it has nearly no convergence in the same setup as DoE and CLUB (Figure 7(a)), as explained in McAllester & Stratos (2020) MINE requires an exponential amount of samples for convergence. We did not consider InfoNCE (van den Oord et al., 2018) since its

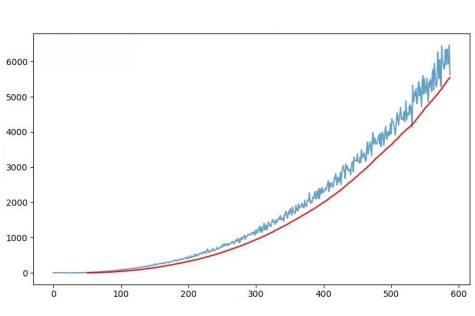

(a) Train data convergence of MINE estimator

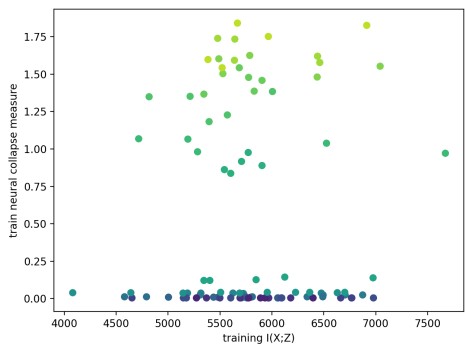

(b) Geometric compression VS MI by MINE

Figure 7: MINE estimator on ResNet18 representations with CIFAR-10 data.

estimation is capped by the batch size used for estimation, and our measured MI can be significantly larger than a realistic batch size we can train the estimator with.

### C.4 DOE HYPER PARAMETERS

In our implementation the encoder $f_\psi(x)$ produces deterministic features $z = f_\psi(x)$, but we use dropout inside the encoder to obtain multiple stochastic views of the same input. For each input $x_i$ we generate

$$z_i^{(1)}, z_i^{(2)}, \ldots, z_i^{(K)}, \tag{7}$$

where each $z_i^{(k)} = f_\psi(x_i; \epsilon_k)$ corresponds to an independent dropout mask $\epsilon_k$. These samples should *not* be treated as independent draws from the marginal $p(z)$; they share the same underlying input and differ only due to injected noise. Consequently, they cannot be used as negative samples against each other.

If the dropout-generated samples $\{z_i^{(k)}\}$ were used as negatives for the same $x_i$, the objective would force the conditional density estimator $q_\theta(z|x)$ to assign low likelihood to dropout variants of the *same* input. This contradicts the intended semantics of $q_\theta(z|x)$, which should assign high density to *all* stochastic realizations of $z$ arising from the same input $x$. In other words, these samples

approximate *samples from the same conditional distribution $p(z|x_i)$* and are therefore all positive examples, not negatives.

For each input $x_i$ we use:

- **Positive set:** all dropout-generated features $\{z_i^{(1)}, \ldots, z_i^{(K)}\}$.
- **Negative set:** dropout samples of *other* inputs, that is

$$\mathcal{N}_i = \{z_j^{(k)} \mid j \neq i, \ k = 1, \ldots, K\}. \tag{8}$$

  These approximate draws from the empirical marginal $p(z)$ and are valid negatives for $x_i$.

Thus, for a minibatch of size $B$ with $K$ dropout samples each, we obtain:

- $K$ positive samples for each of the $B$ inputs;
- $(B-1)K$ negative samples per input.

The logistic DoE objective becomes

$$\mathcal{L}_{\text{cond}} = \sum_{i=1}^{B} \sum_{k=1}^{K} \left[ \log q_\theta(z_i^{(k)}|x_i) - \log \left( \sum_{z \in \mathcal{N}_i \cup \{z_i^{(1)}, \ldots, z_i^{(K)}\}} q_\theta(z|x_i) \right) \right]. \tag{9}$$

Note that all dropout samples of the same input appear in both the numerator (as positives) and in the normalizer of the denominator (to retain the standard noise-contrastive form), but they are never treated as negatives.

This construction preserves the semantics of the conditional model:

- $q_\theta(z|x_i)$ learns to assign high likelihood to all natural stochastic variations of features arising from $x_i$,
- and low likelihood to samples originating from other inputs $x_j$.

Thus the use of dropout increases the effective sample size of $p(z|x)$ without corrupting the negative sampling structure, leading to a more faithful estimation of the conditional entropy.

DoE estimator of conditional entropy was selected depending on the data: For images we employ a simple convolutional architecture and for text a two-layer attention network. We train both estimators with AdamW optimizer with learning rate $1e-4$, batch size $256 * 4$ (with 4 samples from dropout) and gradient clipping to 1. Since convergence of MI estimators is an important characteristic of the quality of the obtained values, we demonstrate here an example of one of the setups measurements. The training data allows for a converged estimation, while test data is close to the convergence, but not very precise. Unfortunately, estimation on one class, needed for conditional MI, cannot converge properly since amount of samples is too small.

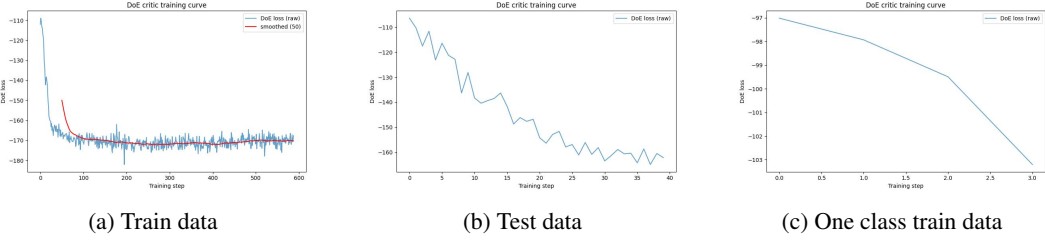

(a) Train data  (b) Test data  (c) One class train data

Figure 8: Convergence of the estimator of $H(Z|X)$ for DoE on ResNet18 representations with CIFAR-10 data.

# D  THEORETICAL ARGUMENT FOR NEGATIVE CORRELATION

We now provide some theoretical arguments that should intuitively explain the observed negative correlation between geometric and information-theoretic compression. To this end, we consider a simplified setting.

We consider a set of points from $\mathcal{X}$ with corresponding classes from $\mathcal{Y}$. We define a stochastic encoder, which maps inputs $X \in \mathcal{X}$ to a two-dimensional latent space $\mathcal{Z}$, encoding a datapoint $x$ with class $y$ to a random variable $Z \sim \mathcal{N}(\mu(x), \sigma^2(x))$, where:

1. $\mu(x)$: The encoder is assumed to distribute all samples of a given class around a learned class centroid. While we assume that this mapping is bijective, we emulate the spread of the encoder output due to differences in the input via drawing from a Gaussian distribution. More specifically, we define the function $\mu$ via drawing, for each datapoint $(x, y)$, a sample from $\mathcal{N}(\mu_y, \sigma_z^2 I)$. Here, $\mu_y$ corresponds to the class centroids, while $\sigma_z^2$ corresponds to the spread of the encoder function $\mu$ (which we assume identical for every class, for the sake of simplicity).

2. $\sigma^2(x) = \sigma_n^2$: The encoder adds noise with a fixed variance to the latent representation.

Note that since the mapping from $X$ to $\mu(X)$ is bijective, its mutual information is infinite, also $I(X; Z) = I(\mu(X); Z)$. Using the formula for the MI of jointly Gaussian random variables, we obtain

$$I(X; Z|Y) = \log\left(1 + \frac{\sigma_z^2}{\sigma_n^2}\right). \tag{10}$$

Hence, large values of $\sigma_n^2$ lead to small values of $I(X; Z)$. For the clustering perspective, small values of $\sigma_z^2$ and $\sigma_n^2$ lead to latent representations that are strongly clustered in a geometric sense.

We simulate this system by drawing the class means $\mu_y$ from $\mathcal{N}(0, I)$ and varying the parameters $\sigma_z^2$ and $\sigma_n^2$ to study their influence on $NC$. To compensate for random effects, we repeat this procedure 50 times.

The resulting measurements are shown in Fig. 1. They demonstrate that information-theoretic compression, i.e., lowering of $I(\mu(X), Z)$, can result from two different causes:

1. A large encoder noise variance $\sigma_n^2$ (lower left corner of Fig. 1(a)), which indicates that noise causes class-specific distributions to overlap;

2. A small variance $\sigma_z^2$ of the encoder mean (upper right corner of Fig. 1(a)), which indicates that samples from the same class are mapped closely in latent space, at least via the deterministic part of $\mu(x)$ of the encoder.

In contrast, the neural collapse measure $NC$ is small only if both $\sigma_n^2$ and $\sigma_z^2$ are small (upper right corner of Fig. 1(b)).

While geometric compression thus aligns well with the concept of tight clusters, information-theoretic compression can have (at least) two causes: geometric compression (in the sense of tight clusters) and uninformative encoders (in the sense of strong encoder noise and/or underfitting). This resonates with the insights of Kolchinsky et al. (2019), and it also explains both the negative correlation between geometric and information-theoretic compression observed in Section 4, and the positive correlation observed in the literature. The correlation is negative if for a fixed spread $\sigma_z^2$ of the deterministic encoder function the added noise increases. For example, in the Conditional Entropy Bottleneck (CEB) framework, the encoder maps inputs $X$ to latent representations $Z$ via a stochastic mapping with additive Gaussian noise, i.e., $Z = \mu(X) + \epsilon$, where $\epsilon \sim \mathcal{N}(0, \sigma^2(X)I)$. Stronger regularization in CEB (via a larger trade-off parameter) encourages the model to reduce the mutual information $I(X; Z)$ by increasing the conditional entropy $H(Z|X)$, which is practically achieved by increasing the variance $\sigma^2(X)$ of the injected noise. This leads to latent representations that are more dispersed or noisier and compress MI.

In contrast, the correlation is positive if, for a fixed noise variance that allows for sufficient MI, the deterministic encoder is varied (as in Goldfeld (2019)).

Since for a trainable stochastic encoder neither $\sigma_n^2$ nor $\sigma_z^2$ can change in isolation, the actual picture will be even more nuanced than described here.

