# OpenReview forum: "Geometric and Information Compression of Representations in Deep Learning"
_ICLR.cc/2026/Conference — Submitted to ICLR 2026_

### Official Review · Reviewer_qMdr · 2025-10-30

**Soundness:** 2
**Presentation:** 2
**Contribution:** 3
**Rating:** 4
**Confidence:** 3

**Summary:**

This work disputes the connection established in previous literature (e.g., by Goldfeld et al., 2019) between the geometric compression of latent representations $Z$ and mutual information $I(X;Z)$, where $X$ are the inputs of a neural network. The authors argue that a decrease in $I(X;Z)$ does not imply a more clustered $Z$; on the contrary, their experimental results suggest that compression correlates with higher mutual information.

The authors also provide a supplementary theoretical result that justifies the Information Bottleneck (IB) analysis of DNNs with analytic activation functions and continuous dropout.

**Strengths:**

The paper's key strengths are its main theoretical result (Theorem 3.1) and the sheer scale of its experimental evaluation. While Theorem 3.1 is an extension of the result from Adilova et al. (2023), I consider it to be important for proving the non-vacuousness of IB analysis for a wider class of neural networks. Furthermore, the experimental results are quite insightful, highlighting the intricate interplay between $I(X;Z∣Y)$, Neural Collapse, various hyperparameters ($\beta$ in CEB and $\lambda$ in the Gaussian dropout framework) and accuracy+generalization.

**Weaknesses:**

I have two major concerns in regard to the methodology:
1. **How MI is measured**.
    - The authors are rather inconsistent with their choice of MI estimators. For Figure 1, they use NPEET (which is, by the way, not referenced in the main text); for CEB, the variational bound is employed; for Gaussian dropout (GD), they use DoE estimator.
    - The paper lacks concrete justification for the choice of estimators. Specifically, the use of NPEET is unexplained, the claim of a "practically tight" variational bound (line 248) is unsubstantiated, and the superior performance of DoE over other SOTA estimators (lines 250-254) is not demonstrated. I kindly ask the authors to elaborate on these decisions and provide the experimental results that support their claims.
2. **When MI is measured.** As stated in lines 128-130, this study focuses on the connection between MI and clustering at the end of training. While this is an interesting direction, I find it only loosely connected to the original works on the IB principle.

    For instance, Shwartz-Ziv & Tishby (2017) measure MI throughout the training. They identified a distinct compression *phase*, where $I(X;Z)$ begins to decrease after a certain epoch. Goldfeld et al. (2019) later connected this *phase* to geometric compression (under normal conditions). Therefore, an anti-correlation between MI and NC observed only at the end of training does not preclude the occurrence of such a compression phase, nor does it rule out geometric compression as its driver (for example, geometric compression can drive MI to the minimal value *throughout the training*, but the minimum itself can still be anti-correlated with NC).

   Moreover, recent studies suggest that compression phases can be transient and may not result in ultimate MI compression (e.g., $I(X;Z)$ might exhibit an overall steady growth punctuated by rapid drops correlated with improvements in training loss). Please, refer to Figure 5 in [1] or non-ReLU IB experiments in [2].

I also encourage the authors to include the full proof of Theorem 3.1, since there is no limit on the length of the Appendix.

[1] Butakov et al. "Information Bottleneck Analysis of Deep Neural Networks via Lossy Compression". Proc. of ICLR 2024.

[2] Anonymous Authors. "A Generalized Information Bottleneck Theory of Deep Learning". ICLR 2025 submission: [https://openreview.net/forum?id=reOA4r0FGL](https://openreview.net/forum?id=reOA4r0FGL).

**Minor issues:**

1. In lines 167-170, the joint distribution of $X$ and $Y$ is said to be "typically continuous", while $Y$ is clearly discrete since the task is classification.
2. The $\parallel$ symbol in Tables 1-2 is not visually appealing due to misalignment. The missing values are also a bit confusing. I understand that they suppose to mean that "gen" requires evaluation on both train and tests subsets. Perhaps, a viable option is placing it in-between the columns using `\makecell` or a similar macro. Finally, it is not immediately obvious what "Perf." stands for. Overall, I suggest an overhaul of these tables.
3. Perhaps, Figure 2 might benefit from log-scaling the `y` axis.
4. The equivalence in line 406 requires additional explanation. As I understand,
   $$
   I(X;Z \mid Y) = I(X;Z) - I(X;Y;Z) = I(X;Z) - I(Z;Y) + \underbrace{I(Z;Y \mid X)}_0,
   $$
   where $I(Z;Y \mid X) = 0$ since $Y \to X \to Z$ is a Markov chain. Please, elaborate on this in the main text.
5. In line 716, a backslash before `log` is missing. I also kindly suggest using `\text` for `dist`, `vol` and `finite` in the subsequent derivations.

**Conclusion:**

Overall, the paper appears rather unpolished. The methodology also needs stronger justification. For these reasons, I recommend a major revision.

**Questions:**

1. Why did you use NPEET instead of DoE for Figure 1?
2. The original implementation of the DoE estimator relies on rather weak approximations of the distributions (e.g., Gaussian). Are they good enough for your complex task?
3. Do you have any intuition behind the anti-correlation between MI and NC? For me, a positive correlation is quite intuitive (clustered representations are "degenerate" and typically encode less information), but I still can not explain the opposite behavior that you observe.

---

> ### Author Response · Authors · 2025-11-20
> **Rebuttal Reply**
>
> We thank the reviewer for a very detailed and thoughtful report. We would first like to clarify **a misconception in the summary: our experimental results do not show that compression correlates with higher mutual information**. Instead, our findings show **negative or non-monotonic correlations between geometric compression and MI**. We appreciate the reviewer’s recognition of **the importance of the theoretical contribution and the breadth of our experimental evaluation**. Below we address all weaknesses and questions.
>
> Weaknesses
>
> 1 Measurement of MI.
> We will improve the clarity of MI estimator usage in the revised manuscript.
>
> - NPEET was used only in the toy example, where the dimensionality is low and the estimator is stable. We now use an analytical formula to evaluate MI.
>
> - In contrast, NPEET is not suitable for the continuous-dropout networks, since it is known to fail in high dimensions. For these cases, we use DoE, which was shown in the original work of McAllester & Stratos (2020) to outperform other high-dimensional MI estimators. Our own preliminary evaluations (not included in the paper) confirmed that DoE exhibits the most stable and consistent measurements among several candidates. We now provide additional experiments with different estimators (see general reply).
>
> - For CEB, MI is already part of the training objective (Fischer et al., 2020), so no external estimator is required.
>
> Importantly, all conclusions in the paper depend on relative/rank comparisons, not absolute MI values. We never mix ranks obtained from different estimators within the same analysis.
>
> 2 Connection to information bottleneck.
> We clarify that our study is explicitly focused on end-of-training representations, not on MI trajectories. Our goal is to test the hypothesis: **“Does low MI at the end of training reliably imply geometrically collapsed representations?”** This is a distinct question from the classical IB trajectory analyses (e.g., Shwartz-Ziv & Tishby 2017), and we do not aim to reproduce or challenge that literature. Goldfeld et al. (2019) also argue that trajectory-based MI analyses may be strongly affected by estimator artifacts. Our focus is instead on the final representation geometry, where the MI value is well-defined and directly comparable across models.
> Even if a network does not undergo MI compression during training (as in Butakov et al. 2024), its final MI is still constrained by the data processing inequality. Thus lower MI in the end of training of one network compared to others leads to the conclusion of MI compression. The reference to an anonymous submission in the current venue is very interesting, but unfortunately addresses completely different information quantities: in particular, using synergy to propose alternative to an information plane analysis based on measuring it.
>
> Regarding the request to include the full proof of Theorem 3.1: the complete proof is already included in Appendix B, or do you suggest to copy Theorem 3.3 from Adilova et al, 2023?
>
> We thank the reviewer for the detailed comments on LaTeX and presentation; these will all be addressed in the revision. To the minor issues:
>
> 1 - The joint distribution is indeed mixed in the case of classification task, but the input distribution is typically continuous.
>
> 4 - Indeed, the equation follows from the Markov chain property. We add this in the text.
>
> Questions
>
> 1 - We previously used NPEET only in the low-dimensional toy example, where it is accurate and stable. Using a closed-form expression for MI, we now evaluate MI for this toy example analytically.
>
> 2 - We use the logistic version of DoE, which is more flexible than the Gaussian variant. Our preliminary tests show that DoE provides the most stable ranking in high dimensions.
>
> 3 - Thank you for this question, we are happy to elaborate on our conjecture. Indeed, the positive correlation is what is more commonly discussed, both experimentally and theoretically (e.g., Sakamoto et al. "Explaining Grokking and Information Bottleneck through Neural Collapse Emergence." ). In this case, and assuming a stochastic network, tight class-specific clustering reduces mutual information, as in the top right corners of Fig. 1. The negative correlation we observe can result from two different phenomena. In the first case, an overly noisy stochastic latent representation becomes entirely uninformative about the task, thus losing cluster structure (large NC1 measure) and simultaneously having small MI. This situation is depicted in the bottom left corners of Fig. 1. In the second case, the stochastic latent representations are tightly clustered, but the (additive or multiplicative) noise does not outweigh the representation’s variation due to the encoder. In such a case, clustering is tight but individual datapoints are still separable leading to large MI. This situation is depicted on the middle of the right edges of Fig. 1 (note that NC measure is plotted logarithmically).

---

### Official Review · Reviewer_gaV2 · 2025-10-31

**Soundness:** 2
**Presentation:** 2
**Contribution:** 2
**Rating:** 4
**Confidence:** 4

**Summary:**

This paper addresses an open question in representation learning: Does low mutual information (MI) between inputs and learned representations imply geometric compression of those representations, and vice versa? The authors probe this through experiments on classification networks with continuous dropout (injecting noise) and with the Conditional Entropy Bottleneck (CEB) objective. They also attempted to examine the role of generalization

**Strengths:**

1) Theoretically sound MI estimation

2) The authors present evidence that one can observe low mutual information without strong within class variation collapse, and variation collapse can occur even when mutual information remains high (as was known for deterministic networks).

3) they also measured that the relationship between generalization and compression is not causal.

**Weaknesses:**

While the experimental design is solid and the question is important, the theoretical framing is not as rigorous.

1) The paper repeatedly refers to “Neural Collapse,” but only measures NC1 (within-class variance). The co-occurrence of NC1-NC2 are critical for a geometry to be called neural collapse (Thm 1 and 2 in papyan 2020). Only NC1 can include degenerative solutions.

2) Neural collapse also refers to when training accuracy goes to 100% (or plateau nearby 100%), did you observe that in your experiments? If not (low beta in the CEB objective, which may lead to compressing away even classification relevant information), it is hard to even say your model attained neural collapse.

**Questions:**

1) Would you please clarify your definition of compression: whether it’s informative compression (e.g., late-phase IB) or trivial compression (e.g., untrained/noisy encoding)?

2) Is it possible to control for test accuracy to demonstrate that generalization is a confounder of compression and low MI between input and latent representations.

---

> ### Author Response · Authors · 2025-11-20
> **Rebuttal Reply**
>
> We thank the reviewer for the thoughtful and invested feedback, and for recognizing both **the importance of the question and the solidity of the experimental design**. We also appreciate **the positive assessment of our theoretically sound MI estimation and the empirical evidence suggesting that information-theoretic and geometric compression do not stand in a simple causal relationship**. Below we address the weaknesses and questions in detail.
>
> Weaknesses
>
> We agree that full neural collapse, in the sense of Papyan et al. (2020), requires the co-occurrence of several properties (NC1–NC4), including classifier geometry alignment and near-zero training error. In this work, our goal is not to claim the emergence of full neural collapse, but rather to study geometric compression through the NC1-based metric introduced by Galanti et al. (2023). This metric captures the tightness of class clusters relative to their separation and is widely used as an operational proxy for geometric compression in empirical studies (Tirer et al. "Perturbation analysis of neural collapse." ICML 2023.; Galanti et al. "On the Role of Neural Collapse in Transfer Learning." CoRR (2021).; Han et al. "Flatness is Necessary, Neural Collapse is Not: Rethinking Generalization via Grokking." NeurIPS 2025.; etc.).
> We will adjust the manuscript to make the terminology precise: (i) We explicitly state that we use NC1-style geometric compression, not full neural collapse. (ii) We clarify that tight class-separated clusters do not imply classifier alignment, but they do quantify the geometric aspect of compression that is central to our comparisons with MI.
>
> Questions
>
> 1. Clarifying our notion of “compression.”
>  A key aim of our work is to disentangle the multiple notions of compression that appear in the literature. For this reason, we intentionally rely on established metrics, NC1 for geometric compression and mutual information for information-theoretic compression, rather than introducing new definitions.
> The toy example in the Appendix D illustrates precisely the reviewer’s point:
>
> - tight clustering (geometric compression) can arise from meaningful abstraction or from degenerate collapse,
>
> - MI can decrease because of improved representations or because of loss of task-relevant information.
>
> This divergence is our main empirical takeaway: **Information-theoretic compression and geometric compression are not equivalent, not even qualitatively monotone, and generalization appears to modulate both in nonlinear ways.** We will emphasize this more clearly in the revised manuscript. We also note that in all our empirical settings we achieve nontrivial training and testing accuracy, so the observed effects are not artifacts of untrained or purely noisy representations.
>
> 2. Generalization as a confounder.
>  We agree that controlling for accuracy to isolate generalization effects would be valuable. In the present work, we treat generalization as a candidate confounder, not a proven causal mechanism. Our results already show that accuracy strongly modulates MI and NC in opposite directions in several regimes (Fig.3), which motivates this hypothesis. A fully controlled causal study with holding accuracy fixed while manipulating geometry and information requires architectural and regularization interventions beyond the scope of this paper. We will clarify this in the discussion and highlight it as an important direction for future work, where we aim to perform proper causal analysis to map the dependency structure among MI, geometry, and generalization.

---

### Official Review · Reviewer_s5LJ · 2025-10-31

**Soundness:** 2
**Presentation:** 3
**Contribution:** 2
**Rating:** 4
**Confidence:** 2

**Summary:**

This paper attempts to better elucidate the relationship between geometric quantities, such as the neural collapse of a neural network, and the information compression that network is capable of. In doing so this paper aims to compare between conditional entropy bottleneck models and models trained with "continuous dropout", and track to what degree neural collapse happens at the same times as information compression.  They find that there are several differences between these different metrics but there is typically a negative relationship between information compression and neural collapse.

**Strengths:**

The proof presented here is to my knowledge novel, and the proof is mathematically interesting and nontrivial. I think that the use of dropout as a way to introduce Stochasticity to allow for the information bottleneck theory to become sensible is also interesting.

The paper also contains a detailed and clear related works section, which can help in reading the literature in this area.

**Weaknesses:**

1. The primary weakness with this paper is that it is not clear, from the empirical results provided, what the takeaway is. Is the takeaway intended to be that neural collapse and information compression are not very strongly or obviously related, as Fig. 3 seems to display? In that case is the purpose of the paper to display a null result (which I think is not an issue, but it should be stated as such)?

**Questions:**

1. From Figure 1, what are we supposed to take from this? My impression is that the neural collapse is somehow orthogonal to the mutual information, is this the right way to interpret this?
2. Are there other ways to measure the mutual information that could provide with more stable estimates for the continuous dropout model?
3. Can you provide some more information as to what the data in Table 1 and 2 are coming from? Are these the combinations of the four considered setups here? Are there differences between them?

---

> ### Author Response · Authors · 2025-11-20
> **Rebuttal Reply**
>
> We thank the reviewer for the positive feedback and **the appreciation of our theoretical contribution and positioning within the literature**. Below we address the weaknesses and questions in detail.
>
> Weakness
>
> Our intended takeaway is the following:
> **Information-theoretic compression and geometric compression are not equivalent,  the qualitative relationship is not even monotonic, and generalization appears to modulate both in nonlinear ways.**
> We will update the manuscript to make this point explicit in the introduction and discussion. This is not a null result; rather, it is a disproof of a common implicit assumption in the compression literature, where it is often taken for granted that  geometry ↔ information ↔ generalization  move in alignment. Our results show that these quantities can diverge substantially, motivating future work to understand the causal relations between different structural properties of learned representations.
>
> Questions
>
> 1. Interpretation of Figure 1.
>  Figure 1 presents the toy example designed to demonstrate that decreases in mutual information and increases in cluster tightness do not need to occur simultaneously. In particular, MI can decrease for reasons unrelated to geometric tightening (e.g., increased noise), whereas NC responds only to geometric structure. We describe these mechanisms in detail in the Appendix D. Our goal is not to claim that the two quantities are “orthogonal,” but to show that they do not reliably coincide and therefore should not be used interchangeably.
>
> 2. Alternative estimators for continuous dropout.
>  Adilova et al. (2023) proposed an MI estimator tailored to continuous dropout. However, our preliminary evaluations indicate that this estimator is not stable in high-dimensional settings and, moreover, it is applicable only to the representation layer immediately following the dropout. For the broader architecture sweep considered in this paper, we therefore opted for DoE, which was demonstrated in its original paper to outperform several baselines and which showed the most stable behavior in our own preliminary consistency checks (not included in the paper). We performed additional calculations with other estimators, to demonstrate that it is not bias of DoE (see general reply).
>
> 3. Clarifying Tables 1 and 2.
>  Table 1 aggregates results across CEB models (MLP, LeNet, WideResNet, DenseNet) and their β-sweeps.
>  Table 2 aggregates results across continuous-dropout models (ResNet-18, VGG-11, DenseNet, MiniBERT) and their λ-sweeps.
> The aggregation procedure is identical for both tables and relies on correlating the normalized ranks of respective target quantities (MI, NC, etc.) as described in Section 4 and AppendixC.1. Differences in the outcomes arise from architectural and dataset differences as well as from underfitting/overfitting behavior, as also illustrated in Figure 3. We will expand the manuscript to describe more clearly what each table summarizes and how the observed patterns align with our broader conclusions.

---

### Official Review · Reviewer_1xgC · 2025-11-01

**Soundness:** 3
**Presentation:** 3
**Contribution:** 2
**Rating:** 4
**Confidence:** 3

**Summary:**

This work examines whether information-theoretic compression, measured by the mutual information I(X; Z), implies geometric compression, quantified by Neural Collapse (NC). The authors find that the relationship is not reliable. Their theoretical and experimental results show that a decrease in mutual information does not necessarily lead to a more collapsed geometric structure.

**Strengths:**

This work establishes the finiteness of mutual information in dropout networks employing analytic activation functions. It presents experiments across different architectures and datasets. The identification of generalization as a potential confounder in the relationship between compression and generalization. The theoretical toy model in Appendix illustrates why mutual information and neural collapse measures can diverge in practice.

**Weaknesses:**

1) The paper relies heavily on the accuracy of MI estimates, yet the justification for the chosen methods is somewhat brief. For CEB, the claim that the variational bound is "practically tight" is asserted but not thoroughly validated. The gap $\mathbb{E} [D_{KL}(p_{Z \mid(|) Y} \mid(|) q_{Z \mid(|) Y})]$ is assumed to be small due to co-training (line 247), but no evidence is provided to quantify this gap.

2) While the DoE estimator might be reasonable choice in some situations, its sensitivity and potential biases in the high-dimensional regimes of state-of-the-art models are not deeply discussed or ablated as far as I know. Thus, it would be more convincing to compare results against a wider suite of MI estimators (line 252).

3) I agree that I(X; Z \mid(|) Y) = I(X; Z) - I(Z; Y), but the claimed I(X; Z \mid(|) Y) \approx I(X; Z) in line 406 should be justified more rigorously.

4) By "geometric compression" the paper means the NC metric. While this is a well-established measure for class-separation geometry and used recently (lines 39-40), it is not the only possible measure. I think the work would be more insightful if it were expanded to include other geometric measures, such as intrinsic dimension, which is mentioned in passing (line 37), or others.

**Questions:**

To further validate the findings, it would be helpful to see results with other MI estimators and to also compute other geometric measures (e.g., intrinsic dimension). This would test whether the correlation with MI holds for geometric properties beyond Neural Collapse.

---

> ### Author Response · Authors · 2025-11-20
> **Rebuttal Reply**
>
> We thank the reviewer for the constructive and encouraging feedback. **We are glad that the theoretical contribution showing that general deterministic neural networks can be endowed with well-defined, finite mutual information via mild noise assumptions and the empirical disentangling of geometric and information-theoretic compression were found valuable.** Below, we address all weaknesses and questions raised by the reviewer.
>
> Weaknesses
>
> 1. Tightness of the CEB variational bound
> We agree that our discussion of the tightness of the variational bound was too brief. As shown by Fischer (2020), the CEB objective jointly trains the encoder and the variational decoder q(Z∣Y)​, and this coordinated optimization explicitly encourages q to match the true class-conditional latent distribution p(Z∣Y). While no variational bound can guarantee perfect approximation of p(Z∣X), the structure of the CEB objective ensures that the KL gap Ep(x,y)​[DKL​(eZ∣X​∥qZ∣Y​)] is actively minimized during training. Moreover, the variational family in CEB, class-conditional Gaussian mixtures, is expressive enough to capture the latent distributions typically observed in CEB models, as also noted in Fischer (2020). Finally, our conclusions rely on rank ordering of MI across models rather than absolute values, further reducing sensitivity to small variational gaps.
>
> 2. Justification for the DoE estimator and estimator bias.
>  We appreciate the reviewer’s point. DoE was chosen because alternative estimators (MINE, kNN-based estimators, etc.) are known to be substantially less stable or significantly more biased in high-dimensional, limited-sample settings (as documented, for example, in McAllester & Stratos 2020). The original DoE paper also demonstrated superior performance relative to existing baselines. In addition, we ran preliminary experiments comparing multiple estimators on our high-dimensional networks; DoE showed the most consistent rankings and stability. We expand Appendix C.3 with a dedicated discussion of these trade-offs, the known limitations of DoE, and explicitly stress that our analysis focuses on relative/rank behavior of MI rather than absolute values. We additionally compare two other estimators in our setup (see general reply).
>
> 3. Approximation I(X;Z∣Y)≈I(X;Z).
>  We agree that this approximation was stated too strongly. For continuous dropout networks, estimating the class-conditional MI I(X;Z∣Y) is challenging in practice: it requires a separate estimate of the distribution p(Z∣Y) for each class. In high-dimensional latent spaces and with limited per-class sample sizes, these class-conditional estimators become extremely unstable. Fortunately, we have: I(X;Z∣Y)=I(X;Z)−I(Z;Y), and the term I(Z;Y) is bounded above by log⁡K, where K is the number of classes. In all our experiments, I(X;Z) is several orders of magnitude larger than log⁡K, and training accuracy is high, so I(Z;Y) is both close to logK in magnitude and stable across models. Therefore, using I(X;Z) rather than I(X;Z∣Y) preserves the relative ordering of models, which is the only information required for our correlation analyses. The conclusions of the paper depend exclusively on this rank structure, not on absolute MI values. We soften the wording accordingly to avoid any confusion.
>
> 4. Choice of Neural Collapse (NC1) as the geometric metric.
>  We appreciate the reviewer’s suggestion to expand to other geometric measures. We focused on NC because it is a widely used and theoretically grounded proxy for class-wise geometric compression, and because it has been explicitly proposed in prior work as being linked to generalization. We will clarify this design choice in Section 3.2. We also explored several alternative geometric measures (Silhouette score and binned entropy of latent codes). Both proved unreliable in our high-dimensional settings due to well-known issues of the curse of dimensionality. Silhouette score relies on absolute L_2​ distances between individual points, but in high dimensions these distances concentrate, and the Silhouette score collapses toward zero regardless of cluster structure. Binned entropy suffers from the exponential growth of bin counts, making entropy estimates extremely unstable and sample-inefficient in high dimension. In contrast, NC1 remains meaningful because it is based on ratios of within-class variance to between-class mean separation. These quantities scale similarly with dimension, so their ratio remains stable. Moreover, NC1 uses class-level statistics (means and covariances) rather than pairwise distances or density estimation, making it far more robust in high-dimensional latent spaces.
> We also note that intrinsic dimensionality is significantly harder to estimate reliably in high dimensions, and even less clear implications for classification settings; Nevertheless, we agree that extending our analysis to additional geometric metrics is an exciting direction for future work.

---

### Author Response · Authors · 2025-11-20
**General Reply to the Reviews**

We thank the reviewers for the encouraging feedback and interesting questions posed. In the following we want to address the most common question raised.

We understand that the community is rightly cautious about mutual information estimation, especially in high-dimensional settings. We would like to emphasize, however, that **our conclusions do not rely on the absolute accuracy of any particular MI estimator**. Throughout the paper, we only use relative (rank) comparisons of MI across models within the same setup. This requires the estimator to be monotonic in the true MI, which is a substantially weaker requirement than absolute fidelity.
To reinforce this and address the reviewers’ concerns directly, we have taken the following steps:
1. **Revised toy example (exact MI)**: We updated the toy example to a fully computational setting where MI can be computed exactly, without any estimator. This eliminates any ambiguity around estimator behavior and illustrates the core phenomenon we highlight: MI and geometric compression can diverge even in the simplest controlled environments.
2. **CEB estimates MI internally**: For CEB models, we do not introduce any external estimator. The MI term in CEB is the variational objective from Fischer (2020), jointly optimized with the encoder and variational decoder. This is the estimator used in the original CEB formulation and does not suffer from the challenges that arise when estimating MI post-hoc. But we will include the estimation with an external estimator if accepted which was impossible due to time constraints during the rebuttal phase.
3. **Gaussian dropout**: For continuous Gaussian dropout models, one must choose a high-dimensional MI estimator. We use the DoE estimator because:

- it was demonstrated in McAllester & Stratos (2020) to be stable and accurate in high dimensions,

- it outperforms its predecessors and avoids the exponential sample complexity issues faced by neural MI estimators.

Since reviewers expressed interest in cross-estimator consistency, we applied two alternative MI estimators to the same trained models in one setup (CIFAR10 with ResNet18). All three produce qualitatively similar ranking. MINE shows less consistency because the estimator fails to converge, which is consistent with the known instability and exponential sample complexity documented in the DoE paper. CLUB behaves well and produces a ranking extremely close to DoE. We also considered InfoNCE, but it is not suitable in our setting because its MI estimate is upper-bounded by the batch size, making it inappropriate for our high-information regime. These additional results are included in the **revised Appendix C.3**.

---

### Meta-Review · Area_Chair_f4Zv · 2026-01-15

**Summary:**

Rejection is recommended. The paper investigates the relationship between information-theoretic compression (Mutual Information) and geometric compression (Neural Collapse) in deep networks. Reviewers (Scores: 4, 4, 4) found the theoretical framing of "geometric compression" via only NC1 (within-class variance) insufficient, as true Neural Collapse involves multiple distinct properties. Furthermore, the reliance on MI estimators in high dimensions raised significant methodological concerns regarding stability and bias, and the empirical results were seen as presenting a null or inconclusive relationship rather than a strong positive finding.

**Reviewer Concerns:**

Despite the rebuttal, key concerns remain outstanding:

Theoretical Framing: The reduction of geometric compression to just NC1 was criticized as incomplete, ignoring the broader definition of Neural Collapse (NC1-NC4).

Methodology: Doubts persist regarding the reliability of MI estimation (DoE) in high-dimensional settings and the lack of rigorous justification for variational bounds.

Interpretation: The core takeaway—that MI and geometric compression are not strongly related—was viewed by some as a null result without sufficient depth or causal explanation regarding generalization.

**Reviewer Scores:**

Scores remained below the acceptance threshold (4, 4, 4). The consensus is that while the question is interesting, the theoretical and methodological limitations prevent the paper from making a definitive contribution.

---

### Decision · Program_Chairs · 2026-01-26

Reject